# Polymers in the Medical Antiviral Front-Line

**DOI:** 10.3390/polym12081727

**Published:** 2020-07-31

**Authors:** Natanel Jarach, Hanna Dodiuk, Samuel Kenig

**Affiliations:** The Department of Polymer Materials Engineering, Pernick Faculty of Engineering, Shenkar College of Engineering and Design, Raman-Gan 52562, Israel; nati.j2@gmail.com (N.J.); Hannad@shenkar.ac.il (H.D.)

**Keywords:** antiviral polymers, drug-delivery polymers, antiviral coating, metal-containing polymers, ionomers

## Abstract

Antiviral polymers are part of a major campaign led by the scientific community in recent years. Facing this most demanding of campaigns, two main approaches have been undertaken by scientists. First, the classic approach involves the development of relatively small molecules having antiviral properties to serve as drugs. The other approach involves searching for polymers with antiviral properties to be used as prescription medications or viral spread prevention measures. This second approach took two distinct directions. The first, using polymers as antiviral drug-delivery systems, taking advantage of their biodegradable properties. The second, using polymers with antiviral properties for on-contact virus elimination, which will be the focus of this review. Anti-viral polymers are obtained by either the addition of small antiviral molecules (such as metal ions) to obtain ion-containing polymers with antiviral properties or the use of polymers composed of an organic backbone and electrically charged moieties like polyanions, such as carboxylate containing polymers, or polycations such as quaternary ammonium containing polymers. Other approaches include moieties hybridized by sulphates, carboxylic acids, or amines and/or combining repeating units with a similar chemical structure to common antiviral drugs. Furthermore, elevated temperatures appear to increase the anti-viral effect of ions and other functional moieties.

## 1. Introduction

Viruses are not living organisms per se, but small structures containing only a nucleic acid genome within a mostly protein based protecting membrane. Unlike living organisms, viruses must penetrate a living host cells to reproduce and replicate [1].

Viruses are usually classified into seven major classes [2,3,4,5,6,7,8]. Class I includes viruses with double-stranded DNA, like Poxvirus, which uses asymmetric transcription to raise their mRNA, much like “regular” cells, but depend on the hosting cell’s polymerases to transcript their genome. Class II contains single-stranded DNA viruses. The DNA’s polarity in these viruses is equal to their mRNA’s. This Class contains the Anelloviridae, Circoviridae, Parvoviridae, Geminiviridae, Microviridae, and more. Class III includes viruses with double-stranded RNA. These viruses’ mRNA demonstrates an asymmetric transcription of the virus’ genome. This group includes the Reoviridae and Birnaviridae viruses. Class IV consists of viruses with single-stranded RNA. These viruses’ mRNA is the sequence’s base identical to their virion RNA. These viruses are referred to as “positive-sense single-stranded RNA viruses”. This group contains Togaviridae, Astroviridae (causing diarrhea), Caliciviridae, Flaviviridae (including many different diseases such as Hepatitis-C, bovine viral diarrhea virus, and more), Picornaviridae (including the rhinoviruses, which causes the “common cold”), Arteriviridae, and Coronaviridae [including the Severe Acute Respiratory Syndrome Coronavirus (SARS-CoV), Middle East Respiratory Syndrome-Related Coronavirus (MERS-CoV), and Severe Acute Respiratory Syndrome Coronavirus 2 (SARS-CoV-2)]. Class V contains viruses with single-stranded RNA. These viruses’ mRNA is complementary in the base to their RNA, and thus also referred to as “negative-sense single-stranded RNA viruses”. This class includes Filoviridae, Arenaviridae, Rhabdoviridae (include the rabies virus), Orthomyxoviridae, Paramyxoviridae, and Bunyaviridae. Class VI contains viruses that along with single-stranded RNA, use the reverse transcription of their positive RNA into a DNA molecule, and thus depend on the host’s polymerases to synthesis their required proteins and to replicate their genome. One example of these viruses is the retroviruses (like HIV). Class VII, one of the smallest classes, consists of viruses with double-stranded DNA that formed into a closed covalent loop. The major example of this group is the Hepatitis-B virus.

The antiviral mechanisms are not always established, and hence common antiviral drug mechanisms are yet to be well understood.

A common antiviral therapeutic method is the use of specific negatively or positively charged particles. Copper and silver ions were of the very first antiviral particles to be studied. In their research about ”the molecular mechanisms of copper and silver ion disinfection of bacteria and viruses”, Thurman and Gerba suggest that positively charged particles (i.e., cations) may affect the DNA/RNA of the viruses [9]. Most microorganisms, including viruses, have some negative charge over their membranes, under near-neutral pH conditions, due to prototrophic groups such as carboxyl, amino, guanidyl, and imidazole which may cause ionization. Cations, therefore, are attracted to the microorganisms’ surface where they might undergo certain reactions. Those cations may also attach to the DNA, RNA, or some enzymes while affecting their action. It should be noted, however, that not all the cations have the same effect – whether it is due to the lack of fitting channels in the membrane or because the cation itself lacks any toxic effect over the microorganism (for example, while copper and silver ions are toxic to most microorganisms, calcium ions are part of their functions and thus lack any toxic effect and might even increase viruses attachment ability [10]). Moreover, as Thurman and Gerba described, neutral particles penetrate the membrane more easily than ions. Thus, neutral complexes (salts) of toxic ions are more effective in the disinfection of bacteria and viruses than pure ions. It should be noted that, as shown by Chambers, the antiviral efficiency of all ions generally increases in proportion to the ion concentration, besides the specific mechanisms that will be described below [11].

While discussing toxic metal ions, several mechanisms have been demonstrated. Lund suggested that the effect of heavy metal ions comes from their oxidation potential, with a higher oxidation potential correlating to a faster reaction [12]. Samuni et al. suggested that due to the “Fenton mechanism”, transition metal ions bind to specific sites in biological polymers/macromolecules. During binding, those ions undergo redox reaction forming secondary radicals next to the original binding target. Then, those radicals attack the macromolecule-metal ion complex, creating H_2_O_2_ which attacks again the complex to form OH radicals. This cyclic redox reaction causes damage to the original molecule and thus leads to its inactivity [13,14]. While this mechanism matches the copper toxic effect [9] and some other metal ions redox reagents [13], three other explanations for the silver toxicity mechanism have been suggested. Tilton and Rosenberg [15] and Khandelwalet et al. [16] suggested that silver might interfere with the essential electron transportation of the microorganisms, and thus cause their elimination. Furthermore, they suggested that silver might bind to DNA/RNA and/or interact with the microorganism’s membrane which leads to functional damage. Petering proposed that while binding to proteins and some other biomaterials, silver form insoluble compounds which is the basic toxic activity of silver and silver ions [17]. Another explanation for the toxicity of both silver and copper ions is that these two metal ions are attracted to the molecules in the cells/viruses more than the cell ions such as phosphate-based ions. When such ions switch positions with the original cell ones, they cause denaturation and/or other structural damage. Moreover, when a metal ion is placed between two particles with hydrogen bonds, a replacement of the hydrogen occurs and then the pH of the surroundings increases [9,18,19]. While these are the major explanations for copper and silver antiviral and antibacterial properties, some other explanations were suggested in the literature, for both these ions and others [9,16,20,21]. Further study conducted by Hashimoto et al. showed that solid-state cuprous compounds, such as cuprous oxide (Cu_2_O), sulphide (Cu_2_S), iodide (CuI), and chloride (CuCl) have high antiviral efficiency, compared to solid-state silver and cupric compounds [22].

Zinc is another example of a metal cation that has been well studied for its in vitro antiviral effect on the viral proteins (at low concentrations) and DNA (at high concentrations) [23]. Read et al. in their review about the role of zinc in antiviral immunity, pointed out that the zinc concentrations that were needed to achieve the antiviral effect (mM [23]) are much higher than the physiological concentrations (μM), with some differences in the needed concentrations between different types of viruses [24]. For example, van Hemert and his associates demonstrated the inhibition effect of some zinc derivatives over the binding and elongation of coronaviruses’ RdRp enzymes [25]. They showed that a combination of 2–320 μM of pyrithione with 2–500 μM of Zn(AC)_2_ has a major effect, with increasing effect at higher concentrations, over SARS-CoV and EAV populations. Moreover, they suggested that water-soluble zinc-ionophores might have a higher antiviral effect. Hong et al. demonstrated the effect of 60–300 μM ZnCl_2_ on the RNA polymerase of Hepatitis C viruses [26]. Takagi et al. also demonstrated the effect of ZnCl_2_, but with lower concentrations of 50–150 μM on the replication of Hepatitis C viruses [27]. Further, 50–200 μM of ZnSO_4_ has an antiviral effect, as shown by Ahlenstiel et al. [28] and by Brendel et al. [23]. Some other Zn salts that showed antiviral effects are pyrrolidine dithiocarbamate (PDTC), zinc gluconate (Zn(Glu)_2_) [24,29], zinc lactate (Zn(Lac)_2_), zinc citrate (CIZAR), zinc picolinate (Zn(pic)_2_), and zinc aspartate (Zn(asp)_2_) [24] as well as zinc ionophores pyrithione [30]. Also, it seems that the antiviral effect of zinc ions is temperature-depended. While at lower temperatures (4–18 °C) zinc has no significant effect on the virus population, at elevated temperatures (20–25 °C) the antiviral effect is significantly increased [23].

While only a few studies have been conducted on the effect of iron ions, it seems that these ions have some level of antiviral effect, as described in Aagripanti et al. work [31]. Cobalt (III) is also a metal ion with antiviral effects [32].

Some other metal-based anions showed antiviral properties like nickel [20], polyoxometalates, polyatomic ions containing three transitional metals such as titanium, vanadium, tungsten, molybdenum, etc. that form clusters with the surrounding oxygen (the transition metal oxyanions linked to each other with sharing oxygen atoms). Those anions show antiviral properties by affecting the virus envelopes/membranes which lead to inhibition of virus infections and an effect on the syncytium formation [33,34].

Non-metal ions have also been studied for their antiviral properties. Pyridinium-based ions, for example, were shown to cause an almost full destruction of viral envelope/membrane, leading leakage of virus DNA/RNA [35]. Quaternary ammonium derivatives are another non-metal cations that have also shown antiviral properties, probably due to their effect on the virus envelopes permeability which is far beyond their viability [36,37,38,39,40,41,42,43,44,45]. This mechanism leads to high efficiency against envelope containing viruses, while its effect on non-envelope viruses is debatable [45]. It should be noted that ammonium derivatives are known to have not only an antiviral effect, but also an antibacterial effect [45,46], and thus form the basis of some commercial disinfectants and cleaners. These commercial products, specifically quaternary ammonium chloride ones, have been tested and found effective against the foot-and-mouth disease virus [47]. Xanthates shows an antiviral effect on RNA and DNA viruses under acidic conditions, as was demonstrated by Sauer et al. in 1987 [48].

Besides ions, other antiviral moieties have been studied. Aurintricarboxylic acid (ATA) could easily be polymerized in water by reacting salicylic acid with formaldehyde, sulphuric acid, and sodium nitrite [49]. Reymen et al. displayed that owing to an interaction between aurintricarboxylic acid analog polymers and the virus membrane/envelope, the replication of DNA and RNA virus was inhibited [49]. Sulphate and phosphorothioate oligonucleotides have been studied for antiviral therapeutics, due to their effect on the hydrophobic properties of the virus envelops, affecting the virus penetrating ability. This effect is amplified with the amount of sulphate or phosphorothioate oligonucleotides moieties, and thus several sulphate/phosphorothioate oligonucleotides containing polymers have been studied, as will be described later [50]. Ganciclovir and Acyclovir are well known commercial antiviral drugs which, though not having any electrical charge, have a well-established effect on some virus polymerase. Some other molecules with antiviral effects are 5-(*N*-ethyl-*N*-isopropyl)amiloride (EIPA), verapamil, and diltiazem. Some studies suggest their effect is a result of the ion-blocking effect. Others suggested that the viral inhibition is based on interactions with virus proteins [51]. Other neutral antiviral molecules are ascorbic and dehydroascorbic acids [52], triclosan [53] and the camphor imine derivatives. Various studies conducted in recent years showed the antiviral potential of the latter, especially against Influenza viruses [54,55,56].

## 2. Polymers for Antiviral Activity

There are two main approaches to the use of polymers in the antiviral campaign. One includes the use of polymers with only a supportive role, using polymers for antiviral drug delivery. The second includes introducing the polymers in the anti-viral front-line using polymers containing ions and/or metal particles or polymers with antiviral moieties such as amines, ions moieties, carboxylic acid moieties, sulphates, phenols and more. Although the first one has been in the focus of most relevant research, only a few examples will be demonstrated related this work, as the second approach is the focus of the current review.
Chapter 1:Chapter 2:

### 2.1. Supporting Role

The enhancement of effective drug release has intensively occupied researchers in recent years. One route suggested is the use of drug-conjugated bio-degradable polymers as drug-delivery agents. By controlling the degradation rate, the drug release could also be controlled. Based on this method, several antiviral-drug-conjugated polymers have been studied over the years. Ahmad et al., for example, studied the uses of a novel chitosan/xanthan gum-based hydrogels as an acyclovir (a commercial antiviral drug) delivery. They found that this cross-linked polymer could be used for antiviral drug delivery, while its efficiency is depended on the drug to polymer ratio, pH, loading time, cross-linking density and other controlled parameters [44].

Other research groups have also studied antiviral drugs that release polymeric systems and mechanisms. Based on their results, it was concluded that the degree of the antiviral effect (i.e., the drug release rate) was dependent on several controllable parameters such as the chemical structure of the polymer, the molecular weight, loading concentrations and the combination with the use of inherently antiviral-polymers [57,58,59]. Liu et al. also studied the effect of temperature on viruses [60]. They found that tobacco plants gained better viral resistance at 30 °C compared to at 20 °C. 

Haam et al. studied the efficiency of using poly(ethylene glycol)-block-poly(phenylalanine) as a drug delivery system, especially against the Influenza A virus [61]. They found that due to the better virus envelope penetrating ability of the polymer compared to the model drugs, the antiviral effect increased. Chen et al. studied the use of poly-l-glutamine (PGN) conjugated with Zanamivir and demonstrated that this system had an antiviral efficiency against early and late-stage Influenza virus infection [62].

Antiviral drug delivery has also been studied using metal-organic frameworks (MOFs), exploiting both their porous structure and their biodegradable properties. An example of this approach is the work done by Gref et al. who studied the use of a MOF obtained by reacting Fe(III) and trimesic acid as an anti-HIV AZT-TP drug delivery system [63]. Other examples are the work of Keskin and Kızılel about the antiviral drug delivery uses of three different MOFs developed by reacting Zn_4_O(CO)_2_ and three different dicarboxylate linkers [64] and the work Novio et al. about the uses of iron–catechol-based MOFs as antiviral drugs delivery systems [65].

### 2.2. In the Front-Line

While polymer antibacterial and antimicrobial properties have been well studied over the years, only a few studies were conducted on the antiviral properties of polymers. Several proposed approaches are discussed in this paper, as follows. First, polymerization of commercial antiviral drugs and/or polymers grafted with commercial drugs (pro)drugs. Second, adding antiviral additives such as metal or ion-particles to achieve an antiviral effect. Other approaches are amine-containing polymers and/or polycations, poly(carboxylic acid)s, polyanhydrides and/or polyanions, sulphate and sulphuric acid-containing polymers and hydroxyl-containing polymers, polyphenols, and/or some other non-ionic polymers. In contrast to their use in a supporting role, described above, when used in a front-line role, polymers do not go through any degradation process to accomplish their antiviral effects.

#### 2.2.1. Polymerization of Commercial Antiviral Drugs [(pro)drugs]

“Macromolecular (pro)drugs”, polymerized commercial antiviral drugs or polymers grafted with commercial drugs as pendant groups, are straightforward examples for polymers with inherent antiviral activity. Several studies have been conducted using this approach, such as Zelikin and his group about Ribavirin [66,67]. Larson also studied the uses of (pro)drugs to overcome immunity against some commercial antiviral drugs. While this approach was found useful in some circumstances, it wasn’t an efficient method for all drugs tested [68,69], thus efficacy might be attributed to other mechanisms.

Bovin and associates, synthesized poly(*N*-2-hydroxyethylacrylamide) (6′SLN-PAA) by polymerization of the co-monomers Neu5Ac*α*2-6Gal*β*1-4GlcNAc*β* (6′SLN) and polyacrylamide as shown in Figure 1 [70]. Owing to the 6′SLN antiviral properties, the entire co-polymer gains antiviral effect against the influenza virus. An increase attraction between viruses and polymer enhanced the drugs’ original antiviral effect.

Based on a similar principle, Carraher et al. studied the effect of two commercial antiviral drugs-based polymers. They found that Cisplatin derivatives containing Tilorone polymers have an antiviral effect on herpes simplex-1, vaccinia, and Varicella zoster and reovirus viruses and a Tilorone derivative polymers have also an antiviral effect on both DNA and RNA viruses [71,72]. Polymers (see Figure 2) were synthesized using Tilorone or Tilorone 11,567 (both commercially available) dissolved in distilled water, and later added to a stirred solution of potassium tetrachloroplatinate(II) in distilled water at room temperature. They also found these two polymers require far lower concentrations to achieve the same antiviral effect, compared with K_2_PtCl_4,_ Acyclovir and/or Cisplatin, as shown in Figure 2.

#### 2.2.2. Antiviral Nanoparticles

The addition of metallic nanoparticles to achieve antiviral properties is one of the most studied approaches. An example is the study conducted by Rashid et al. who examined the addition of silver nanoparticles to polyaniline. They report that this addition contributed to the polymer antimicrobial and antiviral properties [73]. Naka et al. registered a US patent (2007/0169278 A1), based on a similar method, including the addition of metal ions (Ag, Cu, Zn, Al, Mg, and Ca) to textile to obtain an antiviral effect [74]. Imai et al. also explored the addition of copper ions to polymers to obtain an antiviral effect. They showed that when adding copper ions to zeolites (CuZeo), the cotton gained antiviral properties against avian Influenza virus H5 [75]. Silver (Ag) particles have an antiviral effect even in their metal form; Yamamoto et al. showed that the addition of Ag particles to a cotton textile granted it antiviral effect against Influenza A and Feline Calicivirus [76]. Moreover, the addition of Eagle’s minimal essential medium decreased the efficiency of tAg particles. 

Elechiguerra et al. [77] and Lara et al. [78,79] also studied the addition of Ag particles to polymers. They showed that small Ag particles (1–10 nm) added to poly(N-vinyl-2-pyrrolidone), interacted with HIV-1′s gp120 protein causing viral inhibition. Shree et al. found that large Ag particles (~69 nm) interfere with Respiratory syncytial virus cell attachment [80]. Rogers et al. showed that the addition of 10–80 nm Ag particles to acacia gum (a natural polysaccharide) blocked Monkeypox virus interactions with host cells [81]. The same effect on Tacaribe virus was reported by Hussain et al. [82]. Ishihara et al. showed that the antiviral effect of Ag–chitosan composites increase with the concentration of Ag particles [83] and Balagna et al. showed that Ag–silica composite nanoparticles coated masks via a sputtered coating method has an antiviral effect on SARS-CoV-2 virus [84].

Romero and his associates experimented with the addition of zinc acetate (Zn(AC)_2_) to carrageenan-based gels. The combination possessed antiviral properties against herpes simplex virus 2 (HSV-2) and simian immunodeficiency virus (SIVs) both in vitro and in vivo (tested on mises) [85]. 

The anti-viral properties of quaternary ammonium has been studied extensively in the literature [37,38,39]. Lino et al. explored the antimicrobial and antiviral effect of silica nanoparticles coated with didodecyldimethylammonium bromide (DDAB) [86]. Two silica nanoparticles were tested – SNP8 and SNP80, with 8 nm diameter and 80 nm diameter respectively, while the SNP8 was used as a reference. As illustrated in Figure 3, coating glass slides with SNP80 followed by another layer of DDAB rendered the slides with antiviral activity against Influenza A viruses. When they washed the coating, the antiviral effect almost disappeared.

Based on their antiviral effect, some have suggested the use of metallic nanoparticles such as Ag, Mg, TiO_2_ etc. as coating agents to achieve antiviral properties on polymeric surfaces and fabrics. For example, US patent number 2019/0045793 A1 demonstrated TiO_2_, crystalline silver copper aluminum Silicate, alumina and some more metallic-based coatings [87]. The addition of TiO_2_ was also demonstrated by Thilagavathi and Parthasarathi who also studied the addition of metal-based particles to textiles to obtain an antiviral effect aiming at the production of antiviral surgical gowns. It was found that TiO_2_ particles in nonwoven polyester grants an antiviral effect eliminating HIV, Hepatitis B, and Hepatitis C viruses (only) under visible light [88]. For further information about the uses of metal/metal oxide nanoparticles in polymeric matrices to obtain antiviral properties, see the review by Hilal et al. [89].

Another method of antiviral coating was demonstrated by Sahin et al. who studied the consequence of adding 3% sodium pentaborate pentahydrate, 0.03% triclosan and 7% glucapon solution to cotton textile [90]. They reported that after 30 min in solution (and after drying it), the textile gained antimicrobial and antiviral properties. The antimicrobial effect was tested against five bacteria (such as *Escherichia coli, Staphylococcus aureus,* and *Salmonella enterica* subsp), one yeast (*Candida albicans*), two fungi (*Aspergillus niger* and *Trichophyton mentagrophytes*) and two viruses (Adenoid 75 strain of Human Adenovirus type 5 and Chat strain of Human Poliovirus type 1).

#### 2.2.3. Metal-Polymers Hybrids

The use of organo-metal hybrid polymers was also suggested. One example is polymeric anhydride of magnesium and proteic ammonium phspholinoleates that demonstrated antiviral properties against HSV-1, Adenovirus 5 and Canine parvovirus, as shown by Durán and Nunes in their US patent number 5,073,630 [91].

Organotin polymers have also been studied as antiviral polymers. Frank and her group proposed several organotin polymers with antiviral effects against vaccinia virus and Zika virus [92]. Those polymers, as shown in Figure 4, were synthesized using organotin dihalides, camphoric acid and lamivudine. The mechanism for the antiviral effect of organotin polymers is described in the study of Barot et al. During their study, they figured that organotin materials inhibited the cell proliferation and sister chromatid exchanges, which blocked the replication of the cells. Since most viruses depend on the cell replication, these materials have inherent antiviral properties [93]. Some other organo-metalhybrid polymers that have been reported as antiviral polymers are bis(cyclopentadienyl)zirconium dichloride and diethylstilbestrol-based polymers [94].

#### 2.2.4. Amines Containing-Polymers and Polycations

Polyethyleneimine (PEI) is one of the most studied amine-containing polymers in the field of antimicrobial polymers. For example, Vos et al. studied the antiviral effect of branched (PEI) coating over polyether sulphone (PES) membrane [95]. PEI coating led to an increase of -units (99.9%) in MS2 Bacteriophages hosting strain Salmonella Typhimurium WG49 reduction on the membrane surface. As shown in Figure 5, the effect of PEI has been examined on glass slides and by using bulk solution with 1.3% PEI. Besides its inherent antiviral effect, Yang et al. considered some modification of PEI. They showed that owing to their effect on the pH and ability to block viral entry through electrostatic interactions, mannose derivatives containing PEI had antiviral effects [96].

Another study about modified PEIs was conducted by Moeller et al. They examined several modified PEIs obtained by reacting PEI and different cyclic carbonate derivatives [46]. While their study focused on antimicrobial/antibacterial properties, generally, antimicrobial materials also exhibit antiviral properties against enveloped viruses. Modified PEIs were also evaluated by Alvarez-Lorenzo et al. synthesizing modified PEI by polymerization of aziridine and magnetite-silica core-shell particles. It was evident that these polymers inhibited bacteriophage MS2, HSV-1, enveloped viral hemorrhagic septicaemia viruses (VHSV), and nonenveloped infectious pancreatic necrosis virus (IPNV) [97].

Larson studied modified PEI composed of *N,N*-Dodecylmethyl-PEI that exhibited antiviral effect on HSV-1 and HSV-2 viruses (see also Figure 6) [98], influenza A virus [99] and on poliovirus and rotavirus [100]. Other modified PEI tested by Larson was based on *N,N*-hexylmethyl-PEI coated on polyethylene, a surface with antiviral effect eliminating poliovirus and rotavirus [100].

Chitosan has also been studied for its antiviral and antibacterial effects. As been described in Chirkov review, chitosan (and chitin) can induce interferon synthesis, which leads to suppression of the virus replication by causing damage to the RNA and/or mRNA [36]. The chitosan effect was tested on Influenza A, Influenza B, Alfalfa Mosaic virus, bean goldish mosaic virus, peanut stunt virus, TMV, Tobacco Necrosis virus and some other plant viruses. Moreover, some chitosan sulphate derivatives also showed an antiviral effect on HIV-1. Another review about the antiviral effect of chitin polymers (i.e., chitin and chitosan) and their monomer (i.e., *N,N*-diacetylchitobiose dimer and N-acetylglucosamine) was conducted by Rogers et al. deducing similar conclusions [101]. Lei et al. also studied the antiviral effects of chitosan and found that chitosan obtained from Musca domestica L housefl’s larvas has an antiviral effect, based on tests conducted on Autographa californica Multicapsid Nucleopolyhedrovirus (AcMNPV) and Bombyx mori nuclear polyhydrosis virus (BmNPV) [102]. Wu et al. studied a combination of cytosinpeptidemycin and chitosan oligo- saccharide, found it has an antiviral effect on TMS, most likely due to several mechanisms, such as a suppression effect of the viral RNA, an effect on the virus’s subcellular localization as well as punctate formation of TMV MP in some plants leaves [103]. As opposed to the above, Ishihara found chitosan to be ineffective against the Influenza A virus [83].

Lembo et al. studied the effect of other amines-containing polymers like poly(amidoamine)s [104]. Six polymers labeled as ISA1, ISA23, AGMA1, AGMA1_4_, and AGMA1_7_ were studied and found to exhibit antiviral effects on HSV-1, HSV-2, human cytomegalovirus, human papillomavirus-16, respiratory syncytial virus, human rhinovirus, and vesicular stomatitis virus. 

Other amine-containing polymers were synthesized by Xiao et al. by reacting polyhexamethylene guanidine hydrochloride and acrylamide using β-cyclodextrin (CD) with eight active and five active sites. They achieved star polymers with antiviral effect on non-enveloped adenovirus [105]. Pitha et al. researched the antiviral effect of amine-containing polymers. They showed that both poly(9-vinyladenine) and poly(l-vinyluracil) exhibited better antiviral properties, compared to commercial antiviral drugs [106]. 

As forementioned, quaternary ammonium moieties have been studied for their antiviral properties. Duizer et al. investigated the antiviral effect of several hyperbranched quaternary ammonium containing polymers on influenza A (an envelope virus) and poliovirus Sabin1 (a non-envelop virus) [40]. They coated glass and plastic surfaces with the polymers, and then tested their effect on the respective viruses’ populations. To obtain the hyperbranched quaternary ammonium polymers, they reacted in a polymer solution containing polyamine, K_2_CO_3_, tert-amylalcohol and heptylbromide. While the envelope containing Influenza A virus showed significant decay, the non-enveloped Sabin1 showed almost no change.

Klibanov et al. also studied ammonium hybrid polymers for their antiviral properties on Influenza viruses [107]. Polymers labeled 1a–c, 2a–c, 3, 4, and 5 (see Figure 7), exhibited an antiviral effect in correlation with remaining cationic polymers attached to the slide surfaces. It was postulated that the antiviral effect happened by contact between the viruses and the polymers. Moreover, they postulated the polymers labeled as 1a–c and 2a–c possessed probably some other antiviral mechanisms.

Ammonium containing phenylene ethynylene based polymers and oligomers have also been studied as antiviral materials by Whitten et al. [108]. They tested several polymers and oligomers, as shown in Figure 8, against two model non-enveloped viruses, MS2 and T4 bacteriophages. The first is an RNA virus and the latter is a DNA one. They discovered that these hybrid polymers caused a partial dissociation of the virus structure in the dark, while under visible light and/or UV, these materials led to photochemical damage to the viruses’ capsid protein.

Larson investigated the antiviral and antimicrobial effects of ammonium containing *N,N*-dodecylmethyl-polyurethane (synthesis as illustrated in Figure 9) [109]. It was concluded that this polymer, used as a coating, has an antiviral effect against Influenza A viruses (enveloped) but did not affect the non-enveloped poliovirus.

Zhao and associates studied the combined antiviral effect of phosphonium and ammonium, by synthesizing phosphonium-type cationic polyacrylamide as shown in Figure 10 [42]. They showed that this copolymer has an antiviral effect against non-enveloped adenovirus. 

Oligomeric ammonium-silane based systems and their impact on herpesvirus has been studied by Prusty et al. [110]. Their results are consistent with other works related to the antiviral effect of ammonium. 

Shuto et al. patented the antiviral (and antimicrobial) properties of quaternary ammonium ion (US patent number 10,550,274B2), antiviral coatings containing acrylic melamine, quaternary ammonium, multi-valent aromatic carboxylic-acid, and phosphoric acid were included in the claims [111]. It should be noted that the coating was tested against influenza-A viruses.

Quaternary pyridinium containing polymers have also been studied for their antiviral and antimicrobial properties. Xiao and Xue examined the antiviral effect of quaternary pyridinium containing co-polymers on several Influenza viruses (A, PR8, 8, 34), as demonstrated in Figure 11 [35].

As shown by Muñoz-Fernández et al., caprolactam containing polymers also have antiviral properties. They synthesized poly(N-vinyl caprolactam)-nanogels with different levels of cross-linking agents and showed that a polymer with 80 mg of VCL and 4% of BIS crosslinking agent inhibited the replication of R5-HIV-1 viruses in cells [112]. It was concluded that the antiviral effect of this polymer is affected by its thermal properties, collapsing to nanoparticles at body temperatures.

Since copper ions are inherently antiviral, studies have been conducted on polymers conjugated with them. For example, in 2012 Gabbay has registered a US patent consisting of adding copper particles to some polymeric fibers, such as polyamide 6,6 fibers [113]. Moreover, a review on the uses of copper and silver particles has been conducted by Sánchez et al. indicating that antiviral activity can be identified in chitosan with Green seed extract, polyhydroxybutyrate (PHB) with Cinnamaldehyde (for more information about the antiviral effect of cinnamaldehyde, see [114]), poly(lactic acid) (PLA) with silver ions, polyhydroxybutyrate valerate (PHBV) with silver nanoparticles or with copper ions and more [115].

#### 2.2.5. Poly(Carboxylic acid)s, Polyanhydrides and Polyanions

Finkelstein and Merigan studied the antiviral effect of several commercial negatively charged carboxylate-based polymers, as illustrated in Figure 12 [116]. Based on their findings, it was concluded that a higher molecular weight with higher carboxylate free groups led to a higher antiviral effect, although in vivo experiments showed that the polymers were activated only when complexed with organic cations (arginine and poly L-ornithine (PLO)). In addition, bounding the carboxylate groups by amidation reduced the antiviral effect to a minimum. They emphasized that, unlike their previous hypothesis, a higher antiviral effect was achieved using non-degradable polymers.

Some other polycarboxylates were also studied by Loebenstein and Stein [117]. By assessing the antiviral effect of poly(ethylene-co-maleic anhydride), poly(acrylic acid), poly(methacrylic acid), poly(vinyl methyl ether-co-maleic anhydride), poly(vinyl methyl ether-co-maleic acid), poly(vinyl methyl ether-co-maleic anhydride-co-methyl ester), poly(styrene-co-maleic anhydride), poly(isobutylene-co-maleic anhydride) and poly(α-olefin octadecene-co-maleic anhydride) they understood that these co-polymers had an antiviral effect on TMV only in vivo. Regelson and Feltz, who studied the antiviral effect of ethylene-maleic anhydride-based-polymers showed, like Finkelstein and Merigan, that those polymers had an antiviral effect also in vitro [118]. Based on the antiviral effect of the malic acid/anhydride derivatives, Tsunekuni et al. registered a US patent number 2010/0272668A1 claiming several polymeric fibers containing an olefine-maleic acid copolymer, a styrene-maleic acid copolymer, a vinyl ester-maleic acid copolymer, a vinyl acetate-maleic acid copolymer and a vinyl chloride-maleic acid copolymer [119]. Styrene-alt-maleic acid copolymer was also found to inhibit R5 and X4 HIV-1′s infection by Krebs et al. [120]. For further information about the effect of maleic anhydride containing polymers on different viruses, see the manuscript by Popescu et al. [121].

Using the alkyne-azide click chemistry, Mata et al. studied the antiviral effect of carbosilane-containing anionic dendrimers [122]. They found that these negatively charged dendrimers displayed an antiviral effect against HIV-1 viruses and that phosphonate containing dendrimers did not show any antiviral effect.

As a result of the similarity to carboxylate, carboxylic acids containing polymers have been suggested as antiviral polymers, as claimed by Mandeville and Neenan in their US patent number 6,060,235 [123]. Humic acid (HA) based phenolic polymers are polyanion polymers exhibiting antiviral activity. Helbig et al. studied these polymer systems, by reacting and analyzing the effect of different o-diphenolic compounds on the polymers’ antiviral effect on HSV-1 viruses [124]. Their results indicate that increasing the number of carboxylic groups and the number of unsaturated moieties in the starting compounds led to an increase in the antiviral activity of the respective polymers. Other acid-containing polymers reported as antiviral polymers are poly(lysine), poly(glutamic acid), and poly(acrylic acid) [43].

Sano has considered the effect of alginate on TMV [125], which is a natural anionic polymer, usually obtained from seaweeds [126]. He indicated that alginate had an antiviral effect on TMS, demonstrating greater effect with decreasing polymer chains’ stiffness [125]. As shown by Mooney and Lee, alginate also exhibits an antiviral effect on Adenovirus [126].

#### 2.2.6. Sulphate and Sulphuric acid-Containing Polymers

Sulphate containing polymers have also been studied for their antiviral properties. Görög et al. studied the effect of polyvinyl-alcohol-sulphate (PVAS) and polyvinyl-alcohol-sulphate-co-acrylic acid (PAVAS) as inhibitors for HIV-1, HIV-2, Herpesvirus, Human Cytomegalovirus, Vesicular Stomatitis Virus, Respiratory Syncytial virus, Sindbis virus, Semliki Forest virus, Junin virus, Tacaribe virus, and Murine Sarcoma virus [127]. The two polymers were shown to have an antiviral effect over those viruses but do not affect non-envelop viruses such as reovirus and Poliovirus. They also found that the greater the sulphonation degree and/or the greater the molecular weight of the polymers, the higher the antiviral effect became.

Hayashi et al. explored the antiviral effect of modified calcium spirulan, a sulphated polysaccharide. They established that several modified spirulan have an antiviral property. Unlike other studies, non-toxic metal ions have a greater effect compared to copper and silver modified spirulan [128].

Patents have also been registered in the field of sulphate containing polymers, like the US patent of Munson and Tankersley who patented the use of sulphonated naphthalene formaldehyde condensates (SNFC) as antiviral materials [129]. The effect of sulphate on several viruses has also been analyzed by Schelhaas at el. In their inquiry, they tested the antiviral effect of sulphated glycomimetic oligomers and polymers as described in Figure 13 and Table 1 [130]. Spontak et al. studied the antiviral and antimicrobial properties of poly[tert-butylstyrene-*b*-(ethylene-alt-propylene)-*b*-(styrene sulphonate)-*b*-(ethylene-alt-propylene)-b-tert-butylstyrene] (TESET). As already mentioned, they found that the antiviral efficiency of the polymer increases with increasing the degree of sulphonation (in their study, they changed the degree of mol% of the mid-block) [131].

Mimura et al. suggested the use of sulphate modified cellulose and branched cellulose as antiviral polymers. They found that cellulose and branched cellulose with a high degree of sulphonation demonstrated an antiviral effect on HIV-1 [132]. Neurath et al. also pointed the anti-HIV-1 effect of sulphate modified cellulose as well as a similar effect of BufferGel and aryl sulphonates [133].

Sulphated derivatives of natural polymers as antiviral agents have also been demonstrated by Matsuzaki et al. In their research, they studied the antiviral effect of sulphates of curdlan and its branched derivatives. By reacting piperidine N-sulphonic acid or SO_3_-Iimethylformamide complex with curdlan, they obtained antiviral properties against HIV-1 viruses [132]. Hirsch et al. demonstrated an antiviral naphthalene-sulphonate containing polymer with inhibition ability of HIV-1 [134].

The antiviral properties of sulphated polysaccharides have also been studied by Dong et al. In their study, they found that two fucoidans obtained from brown algae *Sargassum Henslowianum* have an antiviral effect on HSV-1 and HSV-2. They suggested the antiviral effect is due to blocking of virion adsorption to host cells [135].

Other antiviral sulphate-containing polymers and their effect on TMV have also been investigated by Sano [136] that showed that two types of polysaccharides, chondroitin Sulphate types C and A, have some inhibition effect on TMV. In a continuing study, he found that the effect of these polymers on the virus might increase or decrease, depending on the molecular weight [137]. Furthermore, the antiviral effect might be attributed to the viral envelope surface protein decapsulation process caused by the polymers.

Combining both amine and sulphate moieties, sulphated cellobiose–polylysine dendrimers were studied by Yoshida et al. [138]. They found that a shorter distance between the terminal sulphate cellobiose units increases the Anti-HIV-1 effect.

Sulphonic acid-containing polymers were studied by Clercq et al. in the course of their study, they synthesized four polymers - poly(4-styrene sulphonic acid) (PSS), poly(anethole sulphonic acid) (PAS), poly(vinyl sulphonic acid) (PVS) and poly(2-acry-lamido-2-methyl-l-propane sulphonic acid) (PAMPS). They concluded these four polymers exhibited antiviral effect on HIV-1 and HIV-2 in MT-4 cells, using concentrations which were not toxic to the hosting cells. They also reported these polymers inhibited the syncytium formation in co-cultures of MOLT-4 cells infected with HIV-1 and HIV-2 [139]. PVS was also tested by Shimonaski et al. with similar results [140]. Cardin et al. examined a sulphonic acid polymer named MDL 101028, a biphenyl disulphonic acid urea copolymer [141]. They found this polymer has an antiviral effect on HIV-1 viruses. Sulphites containing polymers also show an antiviral effect, as was demonstrated by Christopher et al. in their US patent number 20170304815A1 [142]. They demonstrated a change in the oxidation degree of the sulphite’s ions using a catalyst, which led to novel antiviral polymers.

Several hybrid polymers containing anions moieties and sulphate moieties were demonstrated by Orsi et al. [143]. They revealed that dextran sulphate, scleroglucan, and lambda carrageenan have antiviral effects on HSV-1 which is greater than the effect of glyloid Sulphate 4324 and locust bean gum. Furthermore, these polymers influence HSV-2, with similar effect like dextran sulphate, glyloid sulphate 4324 and lambda carrageenan which is higher than the effect of scleroglucan and glycogen sulphate 4435. Finally, they explained that those polymer efficacies were based on their ionic potential.

#### 2.2.7. Hydroxyl-Containing Polymers, Polyphenols and Some Other Non-Ionic Polymers

Polyphenols are another example of antiviral polymers. As was demonstrated by Tempesta in his US patent number 5,211,944, proanthocyanidins-based polymers have antiviral properties against several viruses, including Influenza A, B, and C, respiratory syncytial virus and herpes viruses [144]. Gilbert et al. also confirmed the antiviral effect of polyphenols. They showed an antiviral effect of SP-303, a natural polyphenolic polymer, on Respiratory Syncytial and ParaInfluenza type 3 viruses [145].

Due to the antiviral effect of polyphenols, Lebrun et al. suggested the use of Catechin polyphenol grafted non-woven cellulosic fabrics as bio-based cleaning wipes and filters [146]. In their study, they used laccase enzymatic oxidation to obtain the Catechin grafting over the cellulose-based fabrics. These treated fabrics showed antiviral effect (5-log after 1 h) on T4D Bacteriophage virus of Escherichia coli B.

Panarin et al. [147] synthesized several co-polymers (as shown in Figure 14) by reacting 2-dimethylaminoethyl methacrylate and 2-diethylaminoethyl methacrylate with 2-deoxy-2-methacryalamido-d-glucose and *N*-vinyl-*N*-methylacetamide with 2-dialkylaminoethyl methacrylate units. They established the antiviral properties of these polymers against Influenza A virus subtype H1N1. 

Benzophenones were also studied in the field of antiviral polymers. Sun et al. examined the antiviral effect of benzophenones particles (see Figure 15) on poly(vinyl alcohol-*co*-ethylene)-based membranes [148]. They found that these particles performed under visible light (due to the formation of RNMH·) but could also affect viruses in the dark. However, only non-envelope double-stranded DNA viruses were tested. This conclusion points to the potential of these articles as rechargeable antiviral materials.

In addition, 6′-Fluorinated-aristeromycin analogs were proposed as antiviral materials, owing to their effect on the RNA polymerase (RdRp) and the host cell s-adenosyl-l-homocysteine (SAH) hydrolase. Polymers based on this group have been discussed by Jeong et al. as general antiviral polymers, however, some have only limited effect on few types of viruses. The most efficient of these polymers is illustrated in Figure 16, as well as its antiviral test results [149].

Tong and Sankarakumar suggested the use of imprinted polymers. Accordingly, a one-stage mini-emulsion polymerization, as illustrated in Figure 17, could be used to entrap viruses instead of inhibiting their activity [150]. Notably, this method is virus-specific, depending on the imprinted polymer matching a specific virus.

## 3. Temperature Effect on Antiviral Activity

While most antiviral effects reported in literature stem from materials composition, some report on temperature-dependent antiviral effects. Wout et al. studied short-time pasteurization effects on human milk. The process of heating the milk to 72 °C for 16 s is well effective against bovine viral diarrhea, Hepatitis A and C viruses, HIV-1 and HIV-2, porcine parvovirus, and pseudorabies viruses [151]. While heating above 44 °C may cause damage to most viruses, at lower temperatures, as was shown by Rott and Scholtissek [152] and by Goede et al. [153], the viral activity might increase when temperatures are increased. Similar conclusions were reported by Molla et al. while studying the effect of temperature on Poliovirus formation and RNA synthesis [154]. Yamaya et al., however, showed that, above 37 °C, the Influenza viruses’ replication decrease [155], while Reed showed that for Panonychus-citri’s nonoccluded virus, the biocidal temperature (above which the virus population is damaged) is 46 °C [156]. Chan, et al. showed that for SARS-CoV, the biocidal temperature is <38 °C [157] and Harrison showed that Rothamsted Tobacco Necrosis virus biocidal temperature is even lower than Influenza’s [158]. Therefore, it seems that biocidal temperature differs from virus to virus and that DNA viruses tend to be more stable than RNA viruses [159].

Along with the direct effect of temperatures on some viruses’ populations, some researchers have studied the temperature-dependent antiviral effects of ions. For example, Brendel et al. showed that the effect of Zn^2+^ ions is temperature-dependent increased with increasing temperatures [23]. Sánchez et al. studied the antiviral effects of Ag nanoparticles on norovirus [160]. They tested this effect at two different temperatures, namely 25 and 37 °C, and found an increased antiviral effect at higher temperatures.

Kaufman et al., at their investigation about the thermodynamics of the antiviral effect of ions on the interaction between NS3 protein and single-stranded RNA, also suggested that increasing the temperature may increase the antiviral effect [161]. Bisaillon et al. studied the kinetics of metal ions binding to Hepatitis C’s RNA polymerase and suggested the reaction’s kinetics is temperature-dependent; increasing the temperatures caused accelerated binding [20].

Surfactin is a surfactant used for antiviral and antimicrobial applications. Pauli et al. showed that its antiviral effect on the non-enveloped HSV-1, HSV-2, Suid Herpes Virus type 1 (SHV-1), VSV, SIV, Feline Calicivirus (FCV), and Murine Encephalomyocarditis virus (EMCV) increased with increasing temperatures [162]. For SHV-1, the efficiency increased linearly with increasing temperature up to 30 °C, above which the rate of inhibition was too fast to measure. Hogle et al. showed that the efficiency of drugs that bind to Poliovirus also increases with rising temperatures [163].

Tannins are natural polyphenols, that have been studied for their antimicrobial and antiviral properties. Mileva et al. showed that, much like other antiviral materials, Tannins’ antiviral effects are temperature-dependent, and are more efficient at 37 °C than at room temperature [164].

## 4. Summary and Conclusions

The antiviral battle has been the focus of numerous studies over the years. While originally focusing on small molecules based on antiviral drugs, in recent years, researchers started concentrating on hybrid and composite polymers as a promising approach to the global viral problem. The antiviral campaign is waged in two main approaches. The first employing polymers as drug delivery systems based on biodegradable polymers conjugated with antiviral drugs. In this approach, polymers are used in a supporting role, increasing efficacy and half-life of delivered drugs. The second exploiting hybrid and composite polymers. In this approach, polymers are incorporated with metal and/or metal-ions particles like zinc, silver, copper, zirconium, magnesium, tungsten, and more or otherwise hybridized antiviral polymers based on electrically-charged moieties, are employed, whether anionic or anionic, such as carboxylate and/or organic acids/anhydride, or cationic or cationic such as ammonium, phosphonium, and amines. Some other approaches studied include sulphate containing polymers, phenol/hydroxyl-containing polymers and organometal polymers such as organotin-based polymers. Moreover, some studies have suggested the polymerization of commercial antiviral drugs to achieve a more efficient treatment.

Lastly, it was shown that some antiviral effects are temperature dependent. Most of the reported investigations pointed out that increased temperatures enhance the antiviral impact of ions and the hybrid polymers. Thus, a combination of anti-viral polymers along with the ability to increase temperature may be an additional tool to increase antiviral properties.

## Figures and Tables

**Figure 1 polymers-12-01727-f001:**
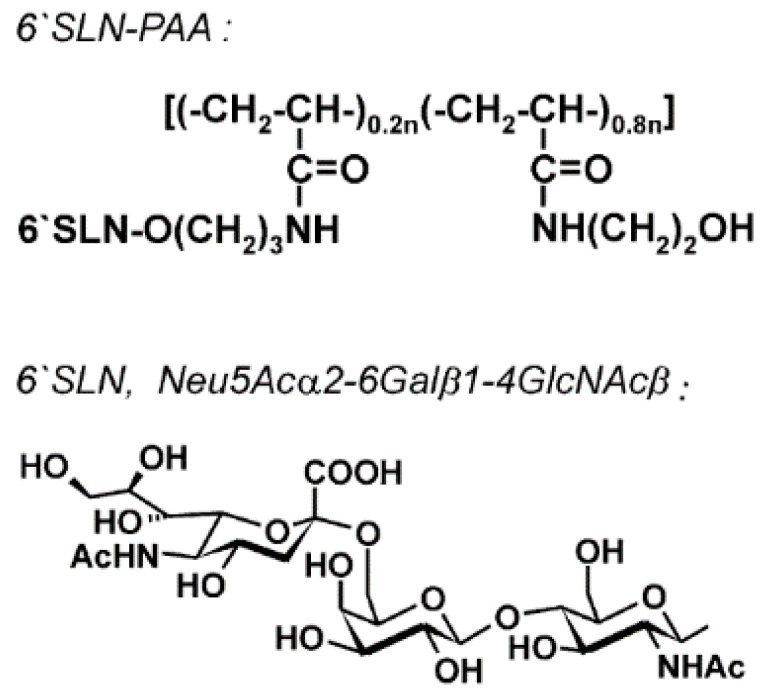
The chemical structure of 6′SLN-PAA and 6′SLN. Reprinted from Gambaryan AS, Boravleva EY, Matrosovich TY, Matrosovich MN, Klenk HD, Moiseeva E V., et al. Polymer-bound 6′ sialyl-*N*-acetyllactosamine protects mice infected by Influenza virus. Antiviral Res 2005, with permission from Elsevier [70].

**Figure 2 polymers-12-01727-f002:**
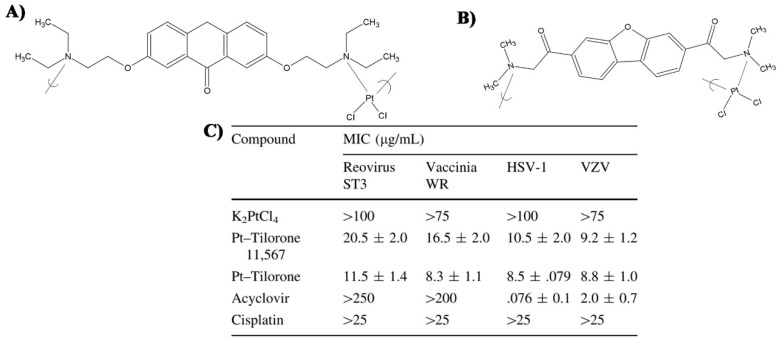
(**A**) Chemical structure of the Tilorone-based polymer; (**B**) Chemical structure of the Tilorone 11,567-based polymer; (**C**) Comparison of antiviral effect of commercial antiviral drugs and new polymers synthesized by Carraher et al., MIC - Minimum Inhibitory Concentration required to reduce virus plaque number by 50%. Reprinted with permission from Roner MR, Carraher CE, Dhanji S, Barot G. Antiviral and anticancer activity of cisplatin derivatives of Tilorone. J Inorg Organomet Polym Mater 2008;18:374–83 [71].

**Figure 3 polymers-12-01727-f003:**
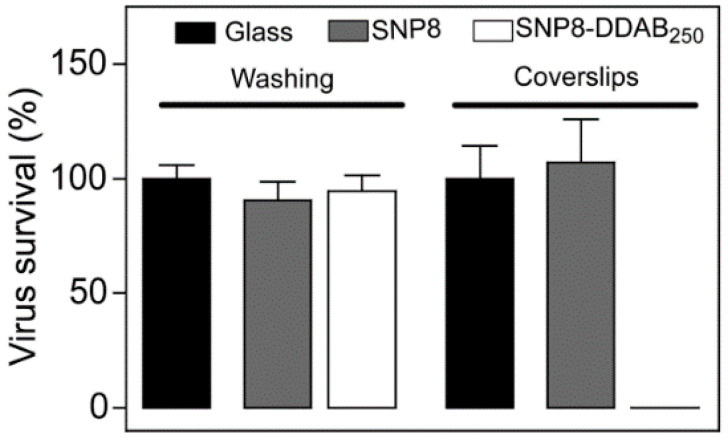
The antiviral effect of SNP80-didodecyldimethylammonium bromide as was tested by Lino et al. Republished with permission from Botequim D, Maia J, Lino MMF, Lopes LMF, Simões PN, Ilharco LM, et al. Nanoparticles and surfaces presenting antifungal, antibacterial and antiviral properties. Langmuir 2012;28:7646–56. Copyright (2012) American Chemical Society [86].

**Figure 4 polymers-12-01727-f004:**
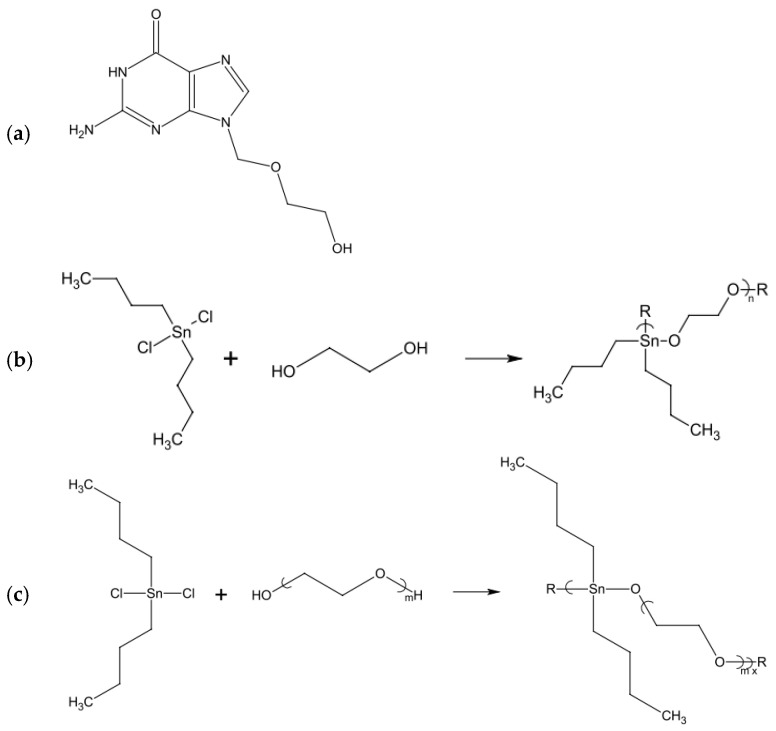
Some of the polymers that were tested by Frank and her group. (**a**)—Structure of polymer repeat unit for the product of acyclovir and diorganotin dihalides; (**b**)—Synthesis of organotin polyethers derived from dibutyltin dichloride and ethylene glycol; (**c**)—Repeat unit for the product of dibutyltin dichloride and poly(ethylene glycol). Reprinted by permission from Springer, Roner MR, Carraher CE, Miller L, Mosca F, Slawek P, Haky JE, et al. Organotin Polymers as Antiviral Agents Including Inhibition of Zika and Vaccinia Viruses. J Inorg Organomet Polym Mater 2020;30:684–94. Copyright (2020) Springer [92].

**Figure 5 polymers-12-01727-f005:**
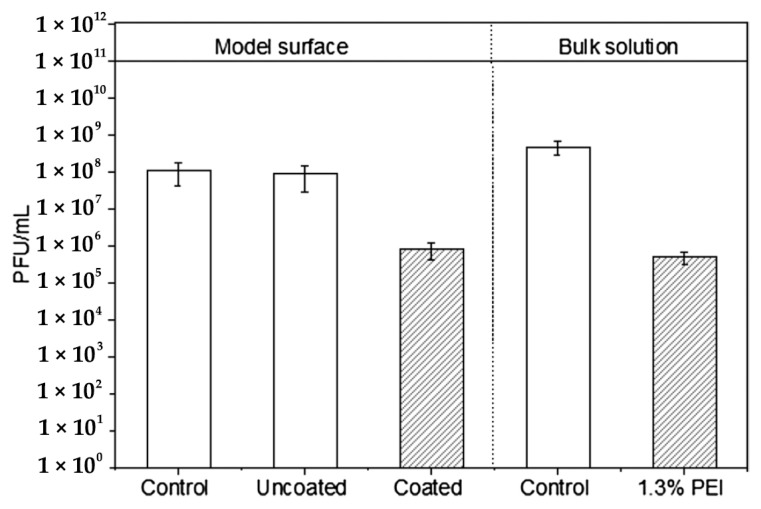
PEI MS2 reduction in model surface (glass-sliders) and in the bulk solution. Reprinted from Sinclair TR, Robles D, Raza B, van den Hengel S, Rutjes SA, de Roda Husman AM, et al. Virus reduction through microfiltration membranes modified with a cationic polymer for drinking water applications. Colloids Surfaces A Physicochem Eng Asp 2018;551:33–41, Copyright (2018), with permission from Elsevier [95].

**Figure 6 polymers-12-01727-f006:**
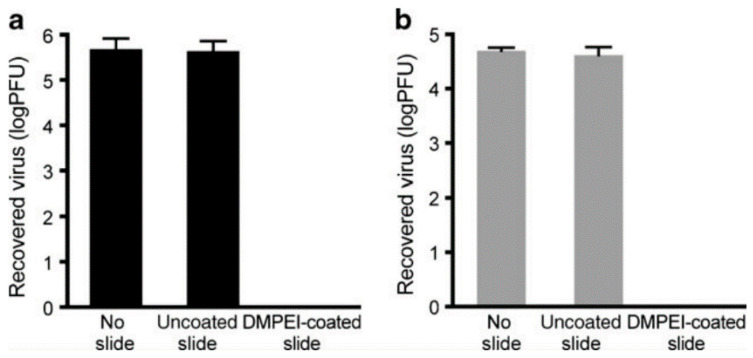
Tests results of *N,N*-Dodecylmethyl-PEI (DMPEI) coating on HSV-1 (**a**) and HSV-2 (**b**). Reprinted by permission from Springer, Larson AM, Oh HS, Knipe DM, Klibanov AM. Decreasing herpes simplex viral infectivity in solution by surface-immobilized and suspended *N,N*-dodecyl,methyl-polyethylenimine. Pharm Res 2013;30:25–31. Copyright (2013) Springer [98].

**Figure 7 polymers-12-01727-f007:**
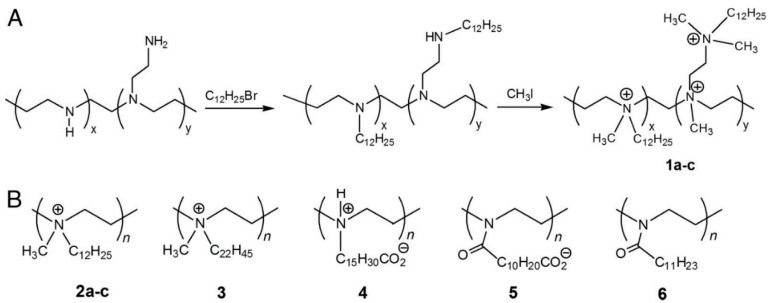
(**A**) Polymerization of 1**a**–**c** polymers and (**B**) chemical structures of polymers 2**a–c**. 3, 4, 5 and 6. Reproduced from Haldar J, An D, De Cienfuegos LÁ, Chen J, Klibanov AM. Polymeric coatings that inactivate both the Influenza virus and pathogenic bacteria. Proc. Natl. Acad. Sci. USA 2006;103:17667–71. Copyright (2006) National Academy of Sciences [107].

**Figure 8 polymers-12-01727-f008:**
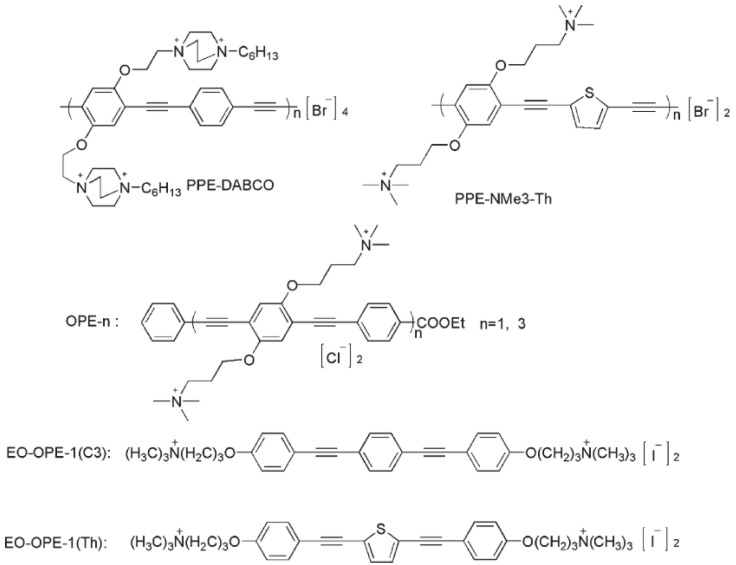
The Ammonium containing phenylene ethynylene based polymers that were tested by Whitten et al. Republished with permission from Wang Y, Canady TD, Zhou Z, Tang Y, Price DN, Bear DG, et al. Cationic phenylene ethynylene polymers and oligomers exhibit efficient antiviral activity. ACS Appl Mater Interfaces 2011;3:2209–14. Copyright (2011) American Chemical Society [108].

**Figure 9 polymers-12-01727-f009:**
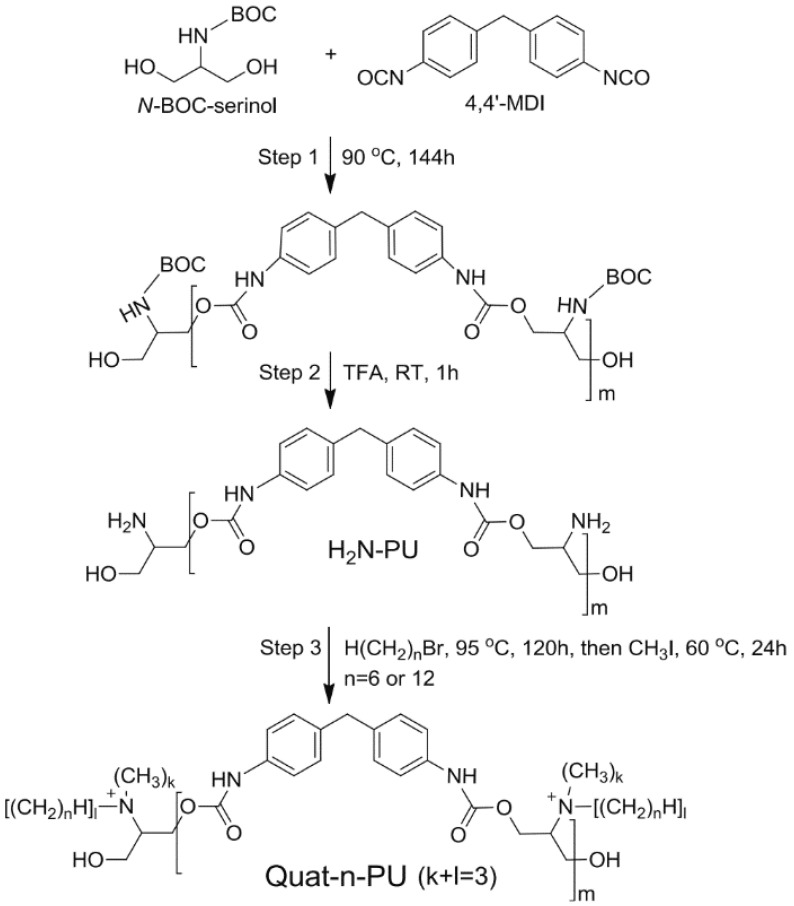
Synthesis of *N*,*N*-dodecylmethyl-polyurethane. Reprinted by permission from Springer, Park D, Larson AM, Klibanov AM, Wang Y. Antiviral and antibacterial polyurethanes of various modalities. Appl Biochem Biotechnol 2013;169:1134–46. Copyright (2013) Springer [109].

**Figure 10 polymers-12-01727-f010:**
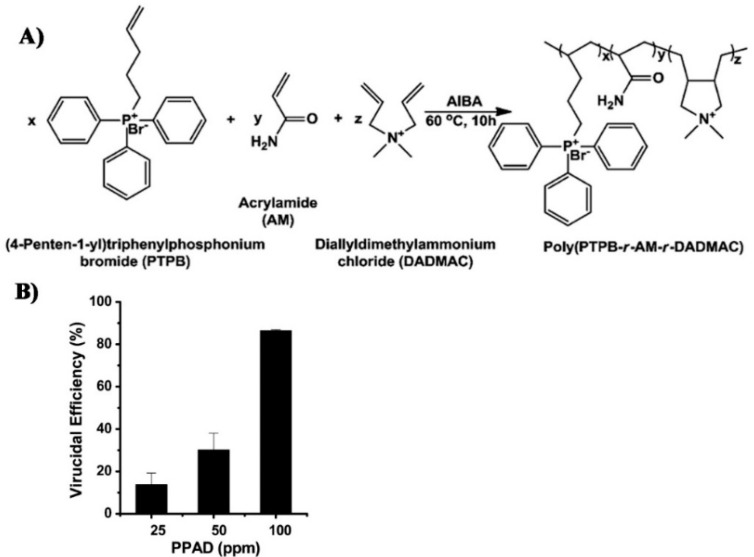
(**A**) The polymers synthesized by Zhao et al.; (**B**) The antiviral effect of the polymer. Republished with permission of the Royal Society of Chemistry (RSC), from Xue Y, Pan Y, Xiao H, Zhao Y. Novel quaternary phosphonium-type cationic polyacrylamide and elucidation of dual-functional antibacterial/antiviral activity. RSC Adv 2014;4:46887–95. Copyright (2014) RSC. Permission conveyed through Copyright Clearance Center, Inc. [42].

**Figure 11 polymers-12-01727-f011:**
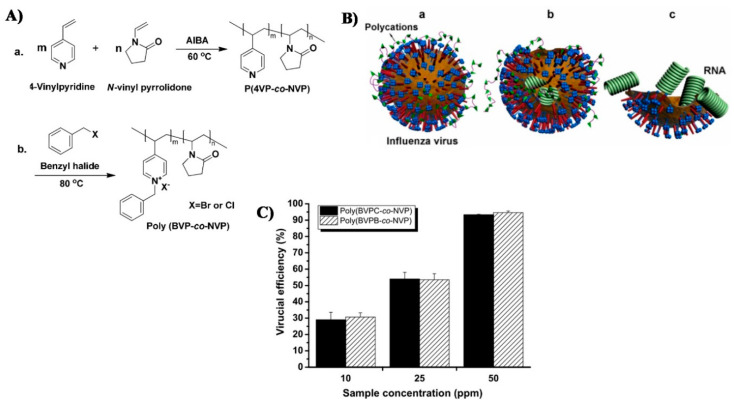
(**A**) The synthesis of the copolymers that were tested by Xiao and Xue; (**B**) The antiviral effect of the quaternary pyridinium containing co-polymers on the Influenza viruses; (**C**) Results of antiviral tests of the two polymers in Influenza viruses, where a is the absorption on the virus’s surface, b is the penetrating of the polymers into the lipid envelope and c is the leakage of the virus’s RNA caused by the damage to the envelope. Based on reference [35].

**Figure 12 polymers-12-01727-f012:**
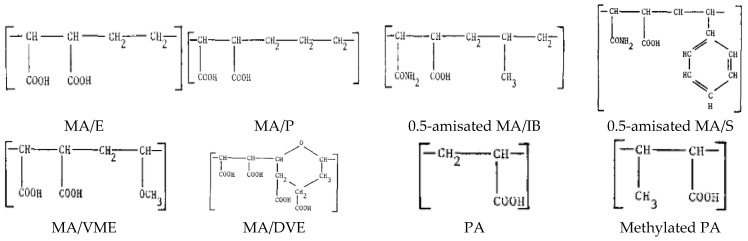
The commercial polymers used by Finkelstein and Merigan. MA—maleic anhydride, DVE—divinyl ether, P—propylene, PA—polyacrylic acid, S—styrene, E—ethylene, IB—isobutylene, VME—vinyl methyl ether. Reprinted from Merigan TC, Finkelstein MS. Interferon-stimulating and in vivo antiviral effects of various synthetic anionic polymers. Virology 1968;35:363–74, with permission from Elsevier [116].

**Figure 13 polymers-12-01727-f013:**
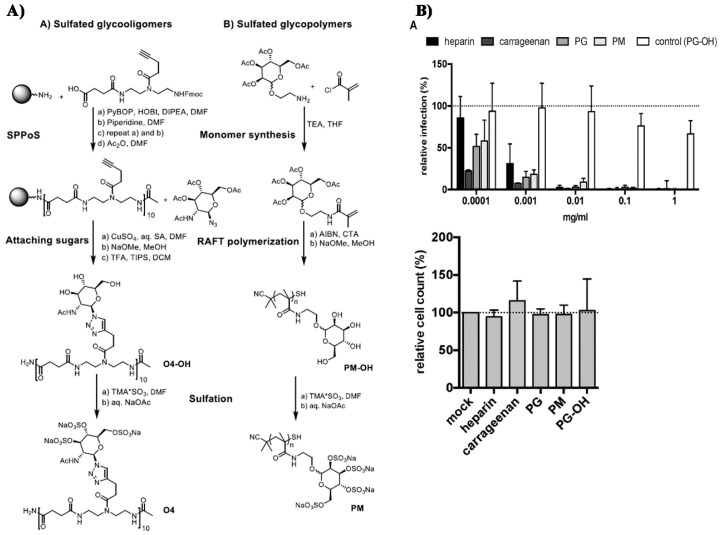
(**A**) General synthesizing of sulphated glycol-oligomers (A) and sulphated glycol-polymers (B); (**B**) Relative cell count and relative cell count of Natural polysaccharides and Sulphated glycopolymers block infection of HPV16. Republished (adapted) with permission from Soria-Martinez L, Bauer S, Giesler M, Schelhaas S, Materlik J, Janus KA, et al. Prophylactic antiviral activity of sulphated glycomimetic oligomers and polymers. J Am Chem Soc 2020. Copyright (2020) American Chemical Society [76].

**Figure 14 polymers-12-01727-f014:**
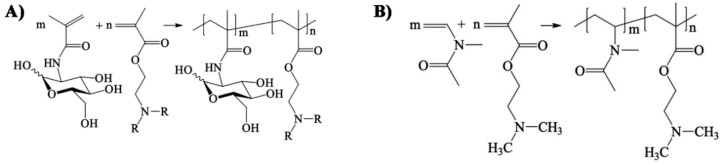
(**A**)—Synthesis of co-polymers by reacting 2-deoxy-2-methacrylamido-d-glucose and DMAEM (R=CH_3_) or DEAEM (R=C_2_H_5_); (**B**)—Synthesis of co-polymer by reacting *N*-vinyl-*N*-methylacetamide and DMAEM. Reprinted by permission from Springer, Nazarova O V., Anan’eva EP, Zarubaev V V., Sinegubova EO, Zolotova YI, Bezrukova MA, et al. Synthesis and Antibacterial and Antiviral Properties of Silver Nanocomposites Based on Water-Soluble 2-Dialkylaminoethyl Methacrylate Copolymers. Pharm Chem J 2020:1–5. Copyright (2020) Springer [147].

**Figure 15 polymers-12-01727-f015:**
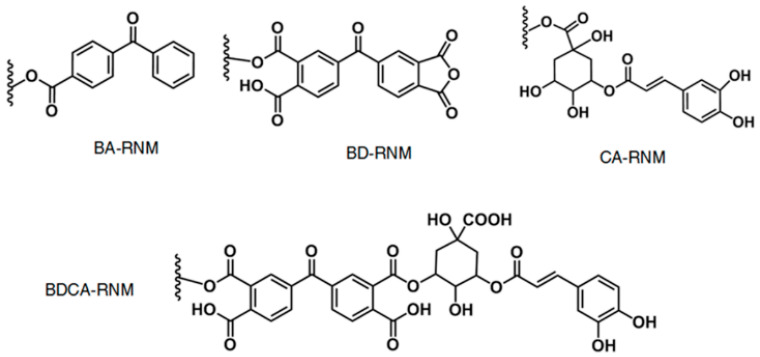
Particles that were tested by Sun et al. Reproduced from Yoon JS, Kim G, Jarhad DB, Kim HR, Shin YS, Qu S, et al. Design, Synthesis, and Anti-RNA Virus Activity of 6′-Fluorinated-Aristeromycin Analogues. J Med Chem 2019;62:6346–62RSC. Reprinted with permission from AAAS [148].

**Figure 16 polymers-12-01727-f016:**
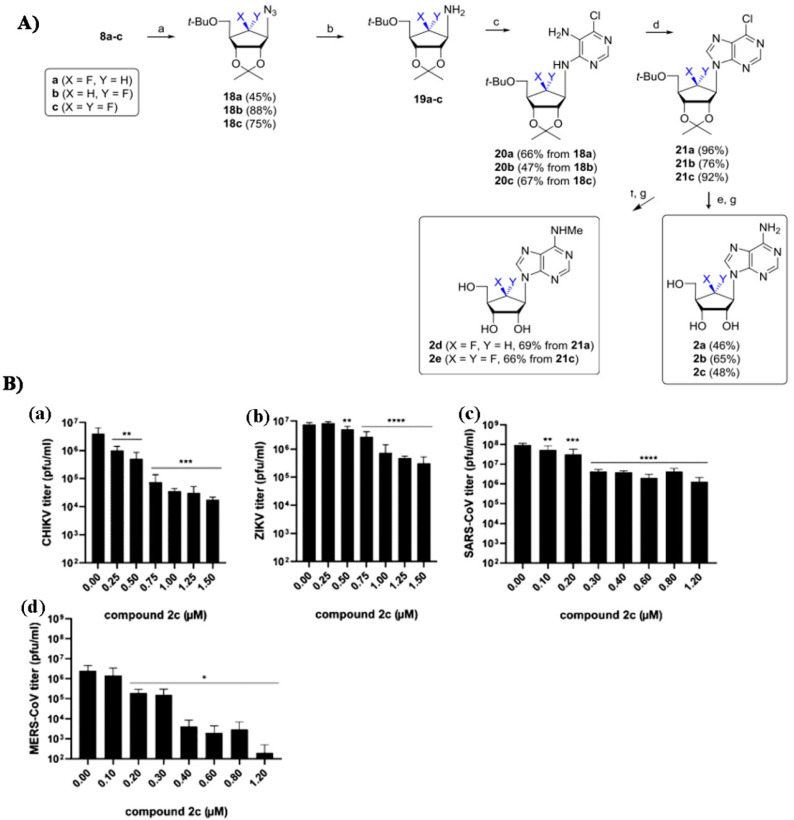
(**A**)—Synthesis of some analogs as was demonstrated by Jeong et al.; (**B**)—Effect of 2c on the infectious progeny of CHIKV (**a**), ZIKV (**b**), SARS-CoV (**c**), and MERS-CoV (**d**). The largest effect was observed on MERS-CoV. Republished with permission Yoon JS, Kim G, Jarhad DB, Kim HR, Shin YS, Qu S, et al. Design, Synthesis, and Anti-RNA Virus Activity of 6′-Fluorinated-Aristeromycin Analogues. J Med Chem 2019;62:6346–62. Copyright (2019) American Chemical Society [149].

**Figure 17 polymers-12-01727-f017:**
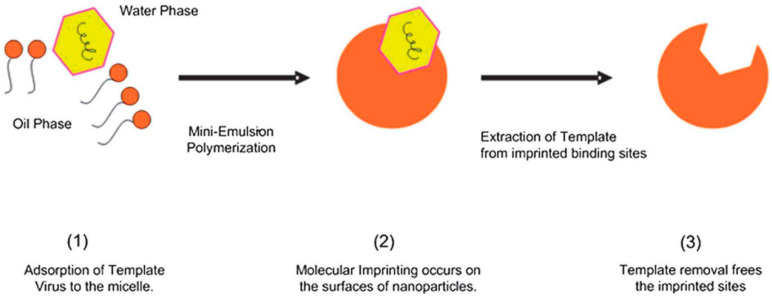
An illustration of virus surface imprinting via one-stage miniemulsion polymerization. Republished with permission of the Royal Society of Chemistry (RSC), from Sankarakumar N, Tong YW. Preventing viral infections with polymeric virus catchers: A novel nanotechnological approach to anti-viral therapy. J Mater Chem B 2013;1:2031–7. Copyright (2013) RSC [150].

**Table 1 polymers-12-01727-t001:** Chemical structure of the sulphated oligomers and polymers that have been studied by Schelhaas at el. Republished (adapted) with permission from Soria-Martinez L, Bauer S, Giesler M, Schelhaas S, Materlik J, Janus KA, et al. Prophylactic antiviral activity of sulphated glycomimetic oligomers and polymers. J Am Chem Soc 2020. Copyright (2020) American Chemical Society [76].

Structure	Dispersity	Degree of Sulphation [%]	N(sugars)	Name
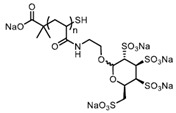	2	n.d	40	PG_1_
See PG_1_	1.08	97.6	46	PG_2_
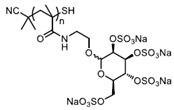	1.25	n.d	86	PM
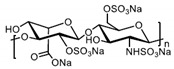	n.a	85/43	30–34	Heparin
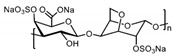	n.a	25-34	n.a	Carageenan
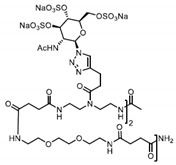	n.a	95–98.5	2	O_1_
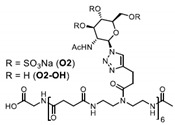	n.a	98.7	6	O_2_
	n.a	0	6	O_2_–OH
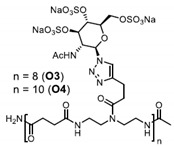	n.a	89.9	8	O_3_
	n.a	85.2	10	O_4_

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
