# Peer review of "Polymers in the Medical Antiviral Front-Line"

_polymers, 2020, doi:10.3390/polym12081727_

Round 1

Reviewer 1 Report

The current manuscript titled "Hybrid Polymers in the Medical Antiviral Front-Line" is a timely contribution to the present pandemic situation. I recommend major revisions to the manuscript based on the comments below.

  1. Title: The title do not reflect the content of the manuscript. Majority of the polymeric systems discussed in the manuscript are either conjugates or composites - very few hybrid polymers?
  2. The figures and chemical structures are not publication quality and should be revised.
  3. The flow of the manuscript is good. Section 2.2. is very long and tedious; and should be divided into subsections. Most the paragraphs start with Author et al. with minimal connection to the preceding content. Section 3 then comes into the scene from nowhere. There is no classification in the manuscript. 
  4. How about adding a section on antiviral polymeric coatings containing metal ions - they are also being used at the frontline?

Author Response

The following is our detailed response to the Reviewers to the proposed manuscript, "Hybrid Polymers in the Medical Antiviral Front-Line". We would like to thank the Reviewers for their meticulous review which upgraded the quality of the paper.

Review 1

The current manuscript titled "Hybrid Polymers in the Medical Antiviral Front-Line" is a timely contribution to the present pandemic situation. I recommend major revisions to the manuscript based on the comments below.

  1. Title: The title do not reflect the content of the manuscript. Majority of the polymeric systems discussed in the manuscript are either conjugates or composites - very few hybrid polymers?

Most of the anti-viral polymers used have a metallic or other inorganic moiety in their structure which make the polymers to hybrid in the molecular or micro levels

  1. The figures and chemical structures are not publication quality and should be revised.

All figures and tables were improved for higher resolution. Some figures were turned into tables, and some were deleted due to copyright issues.

  1. The flow of the manuscript is good. Section 2.2. is very long and tedious; and should be divided into subsections. Most the paragraphs start with Author et al. with minimal connection to the preceding content. Section 3 then comes into the scene from nowhere. There is no classification in the manuscript.

Section 2.2 was split into seven sub-sections. A major editing revision was carried out, with some paragraph merging. A specific mention of section 3 was added to the abstract and the summary.

  1. How about adding a section on antiviral polymeric coatings containing metal ions - they are also being used at the frontline?

Some extra information about coating was added to the relevant sub-section.

We hope that the revision made meet the Reviewers expectation,

Looking forward to receiving your response

Best of health

Prof. Sam Kenig

Reviewer 2 Report

The manuscript “Hybrid Polymers in the Medical Antiviral Front-line” presented by Jarach et al. deals with up-to-date topic – antiviral polymers. There are a lot of important information about hybrid polymers and their antiviral properties. This manuscript is well organized; however, it needs some additional work in order to be suitable for publishing.

  1. There are a lot of problems with figures. Figure 3 is very unreadable, especially the tables. My suggestion is to change the Figure 3 into specific tables, not pictures. Same applies to Figure 4 which is also too big and it almost does not fit the page. Another Figures e.g. 5, 8, 17, 19 have to be improved. They are mostly unreadable and too big. Some change from Figures to Tables has to be made.
  2. Some diagrams and tables should be added for better summary of different topics. This manuscript would benefit from this addition and the topic of this publication would be easier to understand.
  3. There are a lot of editorial issues e.g. “Zn4O(CO)2”. A lot of extra spaces, commas instead of dots. There are also problems with capital letters, which appear when they should not e.g. “poly(vinyl Sulphonic acid)”, “Poly(glutamic acid)”. Moreover, some names of viruses are one time written in capital letters, another time with lower-case letters e.g. Hepatitis, Bovine viral diarrhea viruses. Other mistakes including “the effect of Zn+2 ions” have to be improved also. There are also a lot of word repetition in the abstract and introduction part.
  4. The language has to be checked for all English mistakes e.g. “an envelope virus”, “a non-envelop virus”.
  5. Some more additional info about chitosan and other biopolymers should be added. Chitosan part (lines 376-384) consists of two references. Please elaborate this subsection particularly to justify the applications claims.
  6. The summary and conclusion part is really short, it should be extended and the future part of antiviral hybrid polymers could be added.
  7. More up-to-date references should be added. Also References part should be transformed into correct format suitable for Polymers. It is hard to read the references right now.
  8. Please list all the abbreviations used in the manuscript.

Author Response

The following is our detailed response to the Reviewers to the proposed manuscript, "Hybrid Polymers in the Medical Antiviral Front-Line". We would like to thank the Reviewers for their meticulous review which upgraded the quality of the paper.

Review 2

The manuscript “Hybrid Polymers in the Medical Antiviral Front-line” presented by Jarach et al. deals with up-to-date topic – antiviral polymers. There are a lot of important information about hybrid polymers and their antiviral properties. This manuscript is well organized; however, it needs some additional work in order to be suitable for publishing.

  1. There are a lot of problems with figures. Figure 3 is very unreadable, especially the tables. My suggestion is to change the Figure 3 into specific tables, not pictures. Same applies to Figure 4 which is also too big and it almost does not fit the page. Another Figures e.g. 5, 8, 17, 19 have to be improved. They are mostly unreadable and too big. Some change from Figures to Tables has to be made.

Changed in all figures and tables to a higher resolution was made. Some figures were turned into tables, and some were deleted due to copyright issues.

  1. Some diagrams and tables should be added for better summary of different topics. This manuscript would benefit from this addition and the topic of this publication would be easier to understand.

A major editing was conducted to make the review easier to read and understand.

  1. There are a lot of editorial issues e.g. “Zn4O(CO)2”. A lot of extra spaces, commas instead of dots. There are also problems with capital letters, which appear when they should not e.g. “poly(vinyl Sulphonic acid)”, “Poly(glutamic acid)”. Moreover, some names of viruses are one time written in capital letters, another time with lower-case letters e.g. Hepatitis, Bovine viral diarrhea viruses. Other mistakes including “the effect of Zn+2 ions” have to be improved also. There are also a lot of word repetition in the abstract and introduction part.
  2. The language has to be checked for all English mistakes e.g. “an envelope virus”, “a non-envelop virus”.
  3. Some more additional info about chitosan and other biopolymers should be added. Chitosan part (lines 376-384) consists of two references. Please elaborate this subsection particularly to justify the applications claims.

A major editing revision was conducted, with some paragraph merging, splitting section 2.2 into 7 sub-section and a major English correction.

  1. The summary and conclusion part is really short, it should be extended and the future part of antiviral hybrid polymers could be added.

Additional information was added to the summary section.

  1. More up-to-date references should be added. Also References part should be transformed into correct format suitable for Polymers. It is hard to read the references right now.

More references were added, and all the references' formats were corrected.

  1. Please list all the abbreviations used in the manuscript.

A list of abbreviations was added.

We hope that the revision made meet the Reviewers expectation,

Looking forward to receiving your response

Best of health

Prof. Sam Kenig

Reviewer 3 Report

The review article entitled hybrid Polymers in the Medical Antiviral Front-Line by Jarach et al. provides an overview on polymers as antiviral drugs. In general, this is a timely and interesting review, however manuscript submitted lacks the quality and language to be considered for submission in ‘Polymers’ Journal. Although the content is interesting, the manuscript is full of typos and incomplete sentences and is not readable. I suggest careful revision of language for this article before its resubmission for review purposes. Some more specific comments are given below.
1. Add a graphical abstract to comprehend and overall summary of this paper
2. Redraw figures 5, 14 A, and 17, structures are blurred
3. Redraw tables included in this manuscript, they are very blurred and seem like a copy paste from other papers
4. Consider adding more sub-sections to section 2.0 that will make this easier to read the article
5. The paper is full of typos, incomplete and non-grammatical sentences; some of them are given below but I strongly suggest using native English speaker to revise the manuscript.; some of them are given below but I strongly suggest using native English speaker to revise the manuscript.
A. Rewrite line 29-30, 39-42, 70-73, 165, 169-170, 197-201; these are only a few examples
B. Define term ‘the rate of affection’ on line 77

Author Response

The following is our detailed response (in red) to the Reviewers to the proposed manuscript, "Hybrid Polymers in the Medical Antiviral Front-Line". We would like to thank the Reviewers for their meticulous review which upgraded the quality of the paper.

Review 3

The review article entitled hybrid Polymers in the Medical Antiviral Front-Line by Jarach et al. provides an overview on polymers as antiviral drugs. In general, this is a timely and interesting review, however manuscript submitted lacks the quality and language to be considered for submission in ‘Polymers’ Journal. Although the content is interesting, the manuscript is full of typos and incomplete sentences and is not readable. I suggest careful revision of language for this article before its resubmission for review purposes. Some more specific comments are given below.

  1. Add a graphical abstract to comprehend and overall summary of this paper

A graphical abstract was added.

  1. Redraw figures 5, 14 A, and 17, structures are blurred

Changed in all figures and tables to a higher resolution was made. Some figures were turned into tables, and some were deleted due to copyright issues.

  1. Redraw tables included in this manuscript, they are very blurred and seem like a copy paste from other papers

Changed in all figures and tables to a higher resolution was made. Some figures were turned into tables, and some were deleted due to copyright issues.

  1. Consider adding more sub-sections to section 2.0 that will make this easier to read the article

Section 2.2 was split into seven sub-sections

  1. The paper is full of typos, incomplete and non-grammatical sentences; some of them are given below but I strongly suggest using native English speaker to revise the manuscript.; some of them are given below but I strongly suggest using native English speaker to revise the manuscript.
  2. Rewrite line 29-30, 39-42, 70-73, 165, 169-170, 197-201; these are only a few examples
  3. Define term ‘the rate of affection’ on line 77

A major editing revision was conducted to achieve more flowing reading, with a major English correction.

We hope that the revision made meet the Reviewers expectation,

Looking forward to receiving your response

Best of health

Prof. Sam Kenig

Round 2

Reviewer 1 Report

I feel that the authors did not respond the queries adequately. The biggest concern still remains the flow of the manuscript and the track changes made it worse. I wish I could see a clean copy as well. My comments to authors replies are appended below:

  1. Title: The title do not reflect the content of the manuscript. Majority of the polymeric systems discussed in the manuscript are either conjugates or composites - very few hybrid polymers?

Most of the anti-viral polymers used have a metallic or other inorganic moiety in their structure which make the polymers to hybrid in the molecular or micro levels

Reviewer Comment for R1: I do not agree with this. The authors need to check the definition of the word hybrid to get to the question. For example, two plant species or two animal species can make a hybrid but not 1 plant + 1 animal. Hope this clarifies the issue here.

  1. The figures and chemical structures are not publication quality and should be revised.

All figures and tables were improved for higher resolution. Some figures were turned into tables, and some were deleted due to copyright issues.

Reviewer commmnt for R1: It was inappropriate on part of the authors to not have considered copyright issues from the begining?

  1. The flow of the manuscript is not good. Section 2.2. is very long and tedious; and should be divided into subsections. Most the paragraphs start with Author et al. with minimal connection to the preceding content. Section 3 then comes into the scene from nowhere. There is no classification in the manuscript.

Section 2.2 was split into seven sub-sections. A major editing revision was carried out, with some paragraph merging. A specific mention of section 3 was added to the abstract and the summary.

Reviewer comment on R1: The flow is better than the original but the start of the paragraphs is doesn't comply with the narrative.

  1. How about adding a section on antiviral polymeric coatings containing metal ions - they are also being used at the frontline?

Some extra information about coating was added to the relevant sub-section.

Reviewer comment on R1: The authors should have mentioned where they have made the additions. I couldn't find any substantial information on the same.

Author Response

Point 1: I do not agree with this. The authors need to check the definition of the word hybrid to get to the question. For example, two plant species or two animal species can make a hybrid but not 1 plant + 1 animal. Hope this clarifies the issue here.

Answer to Point 1: The word "Hybrid" was removed from the title and in a few places whee applicable (in yellow)

Point 2: It was inappropriate on part of the authors to not have considered copyright issues from the begining?

Answer to Point 2: The references (3) that were not allowed were  removed from the text. we apologize for the misunderstanding.

Point 3:The flow is better than the original but the start of the paragraphs is doesn't comply with the narrative.

Answer to Point 3: The start of the paragraphs as changed accordingly.

Point 4: The authors should have mentioned where they have made the additions. I couldn't find any substantial information on the same.

Answer to Point 4: The changes were marked in yellow

Reviewer 3 Report

Thank you for carefully revising the manuscript, based upon the reviewer's comments. However, language of the manuscript is still a concern and is not suitable for publication in the current form. I would suggest accepting this article only after major revision to the language and grammar of this article.

Author Response

Point 1: Thank you for carefully revising the manuscript, based upon the reviewer's comments. However, language of the manuscript is still a concern and is not suitable for publication in the current form. I would suggest accepting this article only after major revision to the language and grammar of this article.

Answer to Point 1: The manuscript was checked for language and grammar by online application

Round 3

Reviewer 1 Report

No further comments.